# MM-Prompt: Cross-Modal Prompt Tuning for Continual Visual Question Answering

## Abstract

Continual Visual Question Answering (CVQA) based on pre-trained models (PTMs) has achieved promising progress by leveraging prompt tuning to enable continual multi-modal learning. However, most existing methods adopt cross-modal prompt isolation, constructing visual and textual prompts separately, which exacerbates modality imbalance and leads to degraded performance over time. To tackle this issue, we propose MM-Prompt, a novel framework incorporating cross-modal prompt query and cross-modal prompt recovery. The former enables balanced prompt selection by incorporating cross-modal signals during query formation, while the latter promotes joint prompt reconstruction through iterative cross-modal interactions, guided by an alignment loss to prevent representational drift. Extensive experiments show that MM-Prompt surpasses prior approaches in accuracy and knowledge retention, while maintaining balanced modality engagement throughout continual learning. Our code will be available.

## 1 Introduction

Pre-trained models (PTMs) have achieved considerable success in Visual Question Answering (VQA) (Lin et al., 2022b; Ravi et al., 2023; Li et al., 2021), a traditional multi-modal task that generates answers to questions based on relevant images. Recently, researchers further extended PTMs to Continual VQA (CVQA) (Zhang et al., 2023; Nikandrou et al., 2024; Qian et al., 2023; Lin et al., 2022a; Lei et al., 2023; Cai & Rostami, 2025), where new content emerges over time, with approaches developed to address the challenge of catastrophic forgetting. **Prompt tuning**, a PTM-based method proven effective in continual learning (CL) (Wang et al., 2022a;b; Smith et al., 2023; Menabue et al., 2024), which first selects prompts based on input representations via similarity, then injects the selected prompts into encoders to guide learning. This approach has shown good performance in CVQA (Qian et al., 2023; Zhang et al., 2023; Lei et al., 2023; Cai & Rostami, 2025). For example, Qian et al. (2023) introduces a fusion-prompt pool to complement visual and textual prompts, Lei et al. (2023) replays scene-graph prompts to enhance generalization, and Cai & Rostami (2025) applies K-means clustering to define modality-aware prompt centers.

However, most prompt-based CVQA methods adopt a *cross-modal prompt isolation* approach, constructing and utilizing visual and textual prompts independently without complementary knowledge understanding (Qian et al., 2023; Khattak et al., 2023; Zhang et al., 2023; Cai & Rostami, 2025; Wang et al., 2022a;b; Menabue et al., 2024; Smith et al., 2023; Lei et al., 2023). As illustrated in Fig. 1a, these methods select and process prompts separately for vision and language, without mechanisms for cross-modal interaction. This isolated approach is suboptimal for CVQA, where language features naturally tend to be dominant (Ramakrishnan et al., 2018). Injecting such modality-isolated prompts amplifies the existing imbalance by providing the model with additional information from its already-preferred modality, diminishing the ability to integrate cross-modal representations. As a result, performance degrades as the model increasingly relies on one modality.

To avoid cross-modal prompt isolation and disrupt the error accumulation of modality imbalance in the two stages, i.e., prompt query and injection, we propose MM-Prompt, as shown in Fig. 1b. MM-Prompt consists of two key components, including cross-modal prompt query and cross-modal prompt recovery. First, the *cross-modal prompt query* module enhances prompt selection by mixing information from the opposite modality into each query prior to retrieval, enabling more balanced and context-aware selection. Second, the *cross-modal prompt recovery* module jointly masks and re-

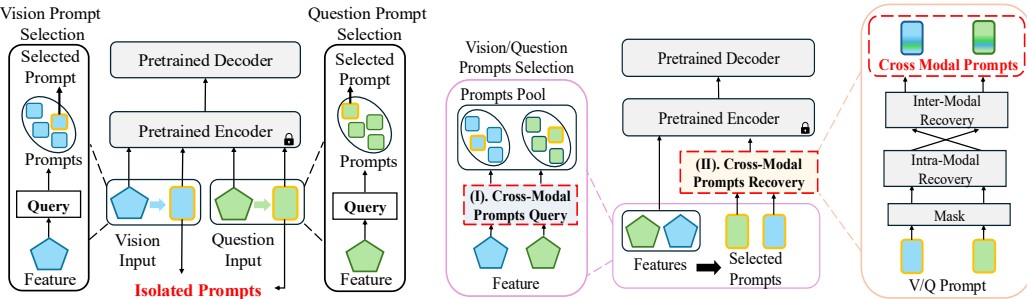

054
055
056
057
058
059
060
061
062
063
064
065
066
067
068
069
070
071
072
073
074
075
076
077
078
079
080
081
082
083
084
085
086
087
088
089
090
091
092
093
094
095
096
097
098
099
100
101
102
103
104
105
106
107

(a) Existing CVQA methods        (b) Our proposed MM-Prompt model

Figure 1: Previous prompt-based CVQA approaches and MM-Prompt. Previous methods rely on isolated prompt usage, accumulating modality-isolated biases, whereas MM-Prompt introduces explicit cross-modal interactions that yield more balanced representations over sequential tasks.

constructs prompts through progressively enriched cross-modal interactions, effectively embedding fused information into the prompt space before the injection stage. Alignment losses are applied to prevent representational drift. Together, these components promote balanced modality utilization, enhance accuracy, and reduce forgetting. Experimental results show that MM-Prompt not only surpasses prior prompt-based methods in performance and retention but also sustains more balanced modality engagement across sequential tasks, effectively addressing the performance degradation caused by modality-isolated prompts that underlies prior failures. Our contributions are:

(1) We show that the cross-modal prompt isolation approach adopted by existing prompt-based CVQA methods progressively amplifies modality imbalance, undermines multimodal reasoning capabilities, and ultimately degrades performance.

(2) We propose Cross-Modal Prompts Query, a mechanism that enriches modality-specific features with complementary information before prompt selection, preventing prompts from reinforcing single-modality specialization

(3) We develop Cross-Modal Prompts Recovery, a masking then hierarchical recovery approach that enforces interactions between modalities and common knowledge understanding between modalities through structured cross-modal pathways

## 2 RELATED WORK

**Continual Visual Question Answering**. CVQA integrates CL approaches by sequentially updating models on new VQA tasks (Qian et al., 2023; Zhang et al., 2023; Cai & Rostami, 2025; Menabue et al., 2024; Smith et al., 2023; Lei et al., 2023). Approaches to address forgetting in CL include regularization-based (Zheng et al., 2023; Lao et al., 2023), architecture-based (Yu et al., 2024b; Li et al., 2019), replay-based (Rolnick et al., 2019; Buzzega et al., 2020), and prompt-based methods (Wang et al., 2022a;b; Smith et al., 2023; Khattak et al., 2023; Menabue et al., 2024; Cai & Rostami, 2025; Zhang et al., 2023; Qian et al., 2023; Lei et al., 2023). While the first three typically require full fine-tuning, prompt-based methods keep backbones frozen and only update lightweight prompts, offering better efficiency. Wang et al. (2022b) selects prompts based on query similarity,Wang et al. (2022a) uses general and expert prompts at different layers and Smith et al. (2023), which injects weighted combinations of prompt components. Despite their success in general CL, prompt methods struggle with CVQA's unique challenges. CVQA suffers from catastrophic forgetting (De Lange et al., 2021) and modality imbalance (Yu et al., 2024a; Peng et al., 2022). Existing prompt methods select and inject prompts in a modality-specific pattern, providing more information to the model's preferred modality, typically language (Yu et al., 2024a), which reinforces modality bias and hinders cross-modal integration. As a result, the visual features are further marginalized, and the model's performance degrades.

**Pre-trained Models with Prompt Tuning in CL**. Pretrained models contain rich, generalizable knowledge for downstream tasks (Han et al., 2021; Raffel et al., 2020; Tan & Bansal, 2019; Douillard et al., 2022; Radford et al., 2021). Among CL approaches, prompt-based methods (Qian et al., 2023; Khattak et al., 2023; Zhang et al., 2023; Cai & Rostami, 2025; Wang et al., 2022a;b; Menabue et al., 2024; Smith et al., 2023; Lei et al., 2023) are the most ideal for pretrained models as they preserve capabilities by keeping backbones frozen. While most prompt methods were developed for unimodal settings, several recent works explored multimodal prompt learning. Qian et al. (2023)

introduced fusion prompt pools with interaction matrices to bridge modalities. Lei et al. (2023) employs scene graphs as symbolic representations to encourage generalized learning across modalities. Cai & Rostami (2025) uses a two-stage process with K-means clustering to establish modality-specific centers before merging them. Despite these advances, these approaches remain suboptimal for CVQA. As learning progresses, modality-specific differences accumulate within prompts without explicit balancing mechanisms. The selected prompts inevitably carry modality-imbalanced information, which can further exacerbate modality disparity and impair cross-modal integration when injected into the model.

## 3 PROPOSED METHOD

### 3.1 PROBLEM FORMULATION

Following Zhang et al. (2023), we formulate CVQA as a sequence of tasks $\{\mathcal{D}_1, \cdots, \mathcal{D}_T\}$. Each task $\mathcal{D}_t = \{(x_i^Q, x_i^V, y_i)\}_{i=1}^{n_t}$ consists of images $x_i^V$, corresponding questions $x_i^Q$, and labels $y_i$. Models must acquire new visual concepts without revisiting past data while preserving performance on earlier tasks. Existing prompt-based methods (Wang et al., 2022a;b) follow a cross-modal prompt isolation strategy, relying on unimodal features for prompt selection and injection, without cross-modal interaction. This reinforces alignment with modality-specific feature distributions, amplifies the dominant modality bias, and hinders the integration of complementary information. To overcome this, we propose MM-Prompt, which introduces two core components: (I) **cross-modal prompt query**, enabling prompt selection guided by fused modality signals, and (II) **cross-modal prompt recovery**, facilitating explicit information exchange across modalities. These components are detailed in Fig. 2.

### 3.2 CROSS-MODAL PROMPT QUERY

In CVQA, vision and language offer complementary representations that benefit from joint modeling (Goyal et al., 2017). Inspired by this, we design a cross-modal querying mechanism that enables each modality to attend to prompts using information from the other. Specifically, we employ an attention (A) module to infuse visual context into question-side queries and linguistic context into image-side queries. For each modality, we maintain a prompt pool consisting of prompts $\mathbf{p} \in \mathcal{P}$ and corresponding keys $\mathbf{k} \in \mathcal{K}$, where $\mathcal{P}$ and $\mathcal{K}$ denote the sets of prompts and keys, respectively, with $\mathbf{p} \in \mathbb{R}^d$ and $\mathbf{k} \in \mathbb{R}^d$. Let $\mathbf{F}^Q \in \mathbb{R}^{l_Q \times d}$ and $\mathbf{F}^V \in \mathbb{R}^{l_V \times d}$ denote the question and vision features extracted from the $x^Q, x^V$, from which we derive the cross-modal queries $\mathbf{q} \in \mathbb{R}^d$ with $\mathbf{w} \in \mathbb{R}^d$:

$$\mathbf{q}^Q = \mathbf{w}^Q \odot \Phi\big(A_{\text{query}}^Q(\mathbf{F}^Q, \mathbf{F}^V, \mathbf{F}^V) + \mathbf{F}^Q\big), \quad \mathbf{q}^V = \mathbf{w}^V \odot \Phi\big(A_{\text{query}}^V(\mathbf{F}^V, \mathbf{F}^Q, \mathbf{F}^Q) + \mathbf{F}^V\big), \quad (1)$$

where $\Phi$ denotes a pooling operation along the sequence. To enable cross-modal integration while preserving modality-specific identity, we enhance the query computation in Eq. (1) with two key components. First, the residual term $(+\mathbf{F}^Q/\mathbf{F}^V)$ retains original modality features during fusion, preventing dilution by cross-modal signals. Second, the weight modulation terms $\mathbf{w}^Q$ and $\mathbf{w}^V$, initialized uniformly and jointly optimized, adaptively control each feature dimension's contribution, emphasizing informative components and suppressing noise. Together, these mechanisms ensure that the enriched queries $\mathbf{q}^Q$ and $\mathbf{q}^V$ integrate complementary cross-modal information without compromising modality fidelity, mitigating uni-modal bias and enabling balanced prompt selection.

Given the enriched queries, we first retrieve the indices for the top-$k$ most similar prompts:

$$\mathcal{I}^M \leftarrow \text{Top-}k\left(\text{sort}\left(\{\cos(\mathbf{q}^M, \mathbf{k}_i^M) \mid \mathbf{k}_i^M \in \mathcal{K}^M\}\right)\right), \quad M \in \{Q, V\}. \quad (2)$$

After getting the indices, instead of relying on the top-1 prompt, which may overfit limited patterns, we compute a weighted aggregation to form more robust cross-modal representations $\tilde{\mathbf{p}} \in \mathbb{R}^d$. $\mathbf{p}_i$ denotes the prompts at $i_{\text{th}}$ indices, the weight $a_i$ is the relevance score:

$$\tilde{\mathbf{p}}^M = \sum_{i \in \mathcal{I}^M} a_i \mathbf{p}_i^M, \quad \text{where} \quad a_i = \frac{\exp(\cos(\mathbf{q}^M, \mathbf{k}_i^M))}{\sum_{j \in \mathcal{I}^M} \exp(\cos(\mathbf{q}^M, \mathbf{k}_j^M))}, \quad M \in \{Q, V\}. \quad (3)$$

By leveraging cross-modal queries, similarity-based weighting favors prompts aligned with joint modality semantics rather than uni-modal features. This promotes the selection of complementary cross-modal cues, effectively mitigating modality bias at the selection stage.

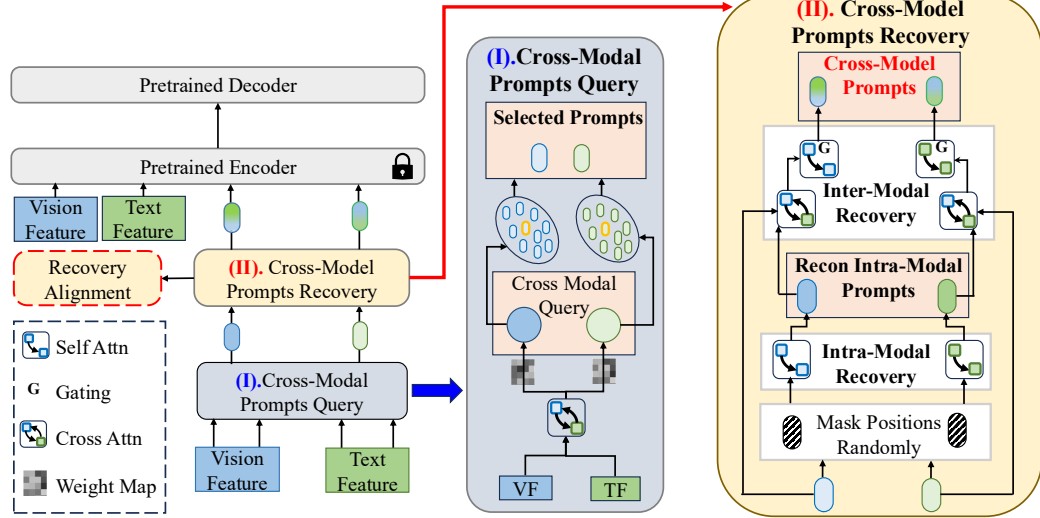

Figure 2: Overall architecture of MM-Prompt. (I).Cross-modal prompts query illustrates our approach where visual (VF) and question (QF) features undergo cross-modal interaction before key matching, creating enriched queries that influence prompt selection across modalities. (II).Cross-modal prompts recovery shows our hierarchical process: identical masks are applied to the prompts from both modalities, followed by intra-modality recovery and inter-modality recovery. This design creates balanced integration pathways that prevent the modality isolation problem in existing prompt-based approaches.

## 3.3 CROSS-MODAL PROMPTS RECOVERY

Existing methods inject prompts directly without further cross-modal interaction, overlooking the integration of complementary knowledge between modalities. This isolated injection worsens modality imbalance by supplying additional modality-specific cues, further biasing the model toward its dominant modality and impairing overall performance. To mitigate this, we propose a cross-modal recovery mechanism, which can be seen in Fig. 2(II).

**Cross-Modal Masking for Alignment and Dependency**. To induce structured interdependence between modalities and encourage explicit cross-modal reasoning, we introduce a shared masking strategy inspired by advances in masked representation learning (He et al., 2022; Zhang et al., 2022). Specifically, we apply identical binary masks to both visual and textual prompt sets, thereby creating aligned missing regions across modalities. These masked regions will be reconstructed using complementary information from the other modality, compelling the model to perform genuine cross-modal integration rather than relying on modality-specific priors. As shown at the bottom of Fig. 2(II), we generate a binary mask $\mathbf{b} \in \mathbb{R}^d$, where each entry is independently set to 0 with probability $\delta$:

$$\hat{\mathbf{p}}^M = \mathbf{b} \odot \tilde{\mathbf{p}}^M, \quad M \in \{Q, V\}. \tag{4}$$

This aligned masking not only enforces balanced modality reliance but also establishes a foundation for hierarchical recovery. Specifically, we leverage the masked prompts in two stages: first, intra-modal recovery restores missing content using internal modality cues, followed by inter-modal recovery, which refines representations through cross-modal interactions. This two-step reconstruction enables progressive integration of modality-specific and complementary information, facilitating deeper cross-modal understanding.

**Intra-Modal Prompt Recovery**. Building on the structured masking, we initiate recovery with a modality-preserving strategy to prevent premature dominance by either modality. Direct cross-modal reconstruction risks overwriting modality-specific patterns or introducing biased alignment. Instead, we adopt an intra-modal recovery phase that reconstructs masked prompts using internal context, augmented with subtle cross-modal signals to establish initial cross-modality awareness. We incorporate an attention module $A(\cdot)$ to reconstruct masked prompts based primarily on their own surrounding context, while a light cross-modal term introduces influence from the opposite modality through a learnable matrix $\mathbf{W}_{res} \in \mathbb{R}^{d \times d}$ initialize with very small values:

$$\mathbf{p}_{intra}^Q = A_{intra}(\hat{\mathbf{p}}^Q) + \mathbf{W}_{res}^Q \tilde{\mathbf{p}}^V, \quad \mathbf{p}_{intra}^V = A_{intra}(\hat{\mathbf{p}}^V) + \mathbf{W}_{res}^V \tilde{\mathbf{p}}^Q. \tag{5}$$

To ensure fidelity and avoid representational collapse, we introduce a dual-objective loss:

$$\mathcal{L}_{\text{intra}} = \sum_{M \in \{Q,V\}} ||\mathbf{p}_{\text{intra}}^M - \tilde{\mathbf{p}}^M||_2^2 + ||\mathbf{W}_{\text{res}}^Q (\mathbf{W}_{\text{res}}^Q)^T - \mathbf{I}||_F^2 + ||\mathbf{W}_{\text{res}}^V (\mathbf{W}_{\text{res}}^V)^T - \mathbf{I}||_F^2, \quad (6)$$

where the $|| \cdot ||_F$ represents the Frobenius norm and $\mathbf{I}$ is an identity matrix. By minimizing this loss, the intra-modal recovery establishes a foundation that preserves essential modality-specific patterns while introducing initial cross-modal awareness, creating balanced representations that serve as the basis for deeper integration. However, it still remains insufficient for modeling richer interactions that require explicit cross-modality reasoning. To address this, we introduce a dedicated inter-modal prompt recovery mechanism that operates on the recovered representations to achieve deeper integration and complete the cross-modal reconstruction process.

**Inter-Modal Prompt Recovery**. Building upon the intra-modal recovery foundation, we next introduce explicit inter-modal integration to enable fine-grained cross-modal reasoning. To this end, we apply attention modules across modalities, allowing each modality's prompts to selectively incorporate complementary features from the other. Residual connections ensure preservation of core modality-specific characteristics during this integration:

$$\mathbf{p}_{\text{inter}}^Q = \mathbf{A}_{\text{inter}}^Q(\mathbf{p}_{\text{intra}}^Q, \tilde{\mathbf{p}}^V, \tilde{\mathbf{p}}^V) + \mathbf{p}_{\text{intra}}^Q, \quad \mathbf{p}_{\text{inter}}^V = \mathbf{A}_{\text{inter}}^V(\mathbf{p}_{\text{intra}}^V, \tilde{\mathbf{p}}^Q, \tilde{\mathbf{p}}^Q) + \mathbf{p}_{\text{intra}}^V. \quad (7)$$

Eq. (7) completes coarse-level inter-modal fusion, enabling the model to capture salient cross-modal signals. However, it may still miss fine-grained or subtle cues critical for downstream reasoning. To address this, we incorporate an additional attention module with twice feed-forward dimension capacity for refinement, but apply it selectively through a learnable gating mechanism that determines where further integration is beneficial:

$$\mathbf{p}_{\text{final}}^Q = (1 - \mathbf{g}^Q) \odot \mathbf{p}_{\text{inter}}^Q + \mathbf{g}^Q \odot \mathbf{A}_{\text{gate}}^Q(\mathbf{p}_{\text{inter}}^Q), \mathbf{p}_{\text{final}}^V = (1 - \mathbf{g}^V) \odot \mathbf{p}_{\text{inter}}^V + \mathbf{g}^V \odot \mathbf{A}_{\text{gate}}^V(\mathbf{p}_{\text{inter}}^V), \quad (8)$$

where the gating mechanism calculates region-specific enhancement weights:

$$\mathbf{g}^Q = \text{Sigmoid}\left(\mathbf{W}_g^Q\left(\left[\mathbf{p}_{\text{inter}}^Q; \tilde{\mathbf{p}}^V\right]\right)\right), \quad \mathbf{g}^V = \text{Sigmoid}\left(\mathbf{W}_g^V\left(\left[\mathbf{p}_{\text{inter}}^V; \tilde{\mathbf{p}}^Q\right]\right)\right).$$

Here, $[\cdot; \cdot]$ represents the concatenation along the feature dimension and $\mathbf{W}_g \in \mathbb{R}^{d \times 2d}$ is a learnable weight matrix that allows the model to adaptively select the areas for enhancement. This design ensures refinement occurs only when supported by strong cross-modal evidence, thus avoiding unnecessary interference and preserving representational sparsity.

To improve inter-modal consistency, we introduce a cross-modal alignment loss that enforces semantic agreement between dual perspectives—how question features are contextualized by vision and vice versa (the numerator):

$$\mathcal{L}_{\text{inter}} = 1 - \frac{\mathbf{A}_{\text{loss}}^Q(\mathbf{p}_{\text{final}}^Q, \tilde{\mathbf{p}}^V, \tilde{\mathbf{p}}^V) \cdot \mathbf{A}_{\text{loss}}^V(\mathbf{p}_{\text{final}}^V, \tilde{\mathbf{p}}^Q, \tilde{\mathbf{p}}^Q)}{||\mathbf{A}_{\text{loss}}^Q(\mathbf{p}_{\text{final}}^Q, \tilde{\mathbf{p}}^V, \tilde{\mathbf{p}}^V)||_2 \cdot ||\mathbf{A}_{\text{loss}}^V(\mathbf{p}_{\text{final}}^V, \tilde{\mathbf{p}}^Q, \tilde{\mathbf{p}}^Q)||_2}, \quad (9)$$

where denominator normalizes both scores to focus on directional agreement rather than magnitude. By minimizing this loss, we encourage bidirectional semantic coherence and prevent representational drift as the model adapts to new tasks.

In summary, inter-modal recovery completes the masked prompt reconstruction pipeline by explicitly integrating cross-modal cues through attention, selective enhancement, and alignment. This phase ensures that final prompt representations are not only modality-aware but also deeply fused, facilitating robust and balanced multimodal reasoning.

## 3.4 TRAINING OBJECTIVE

Finally, these cross-modal prompts are then injected into the frozen pretrained encoder layers. This entire process is trained end-to-end with a multi-component objective:

$$\mathcal{L}_{\text{total}} = \mathcal{L}_{\text{CE}} + \mathcal{L}_{\text{qk-align}} + \alpha \mathcal{L}_{\text{inter}} + \beta \mathcal{L}_{\text{intra}}, \quad (10)$$

where $\mathcal{L}_{\text{CE}}$ is the cross entropy loss that optimizes task performance. $\mathcal{L}_{\text{qk-align}} = \sum_{i \in \mathcal{I}^M}^{M \in \{Q,V\}}(1 - \cos(\mathbf{q}^M, \mathbf{k}_i^M))$ maintains structural quality of our representations through query-key alignment

(Wang et al., 2022a;b). $\mathcal{L}_{\text{inter}}$ and $\mathcal{L}_{\text{intra}}$ ensure consistency between recovered representations during both intra-modal recovery and inter-modal recovery phases. This multi-component loss jointly optimizes for predictive accuracy, cross-modal alignment, and recovery quality, creating a balanced optimization objective that effectively enhances the quality and utility of cross-modal prompts.

As illustrated in Fig. 1, our MM-Prompt model differs fundamentally from existing CVQA approaches by introducing explicit cross-modal interactions at multiple stages. The overall workflow, shown in Fig. 2, consists of two complementary mechanisms that work together to maintain balanced modality representation. First, input features from vision and question modalities ($\mathbf{F}^{\text{V}}$ and $\mathbf{F}^{\text{Q}}$) undergo Cross-Modal Prompts Query. In this stage, each modality's features attend to the opposite modality before generating query vectors. These enriched queries are then used to select relevant prompts. Then, a mask will be applied on the weighted sum of these selected prompts to create explicit pathways for cross-modal prompts recovery. The masked prompts then first process through intra-modal recovery to establish basic modality-specific patterns while introducing light cross-modal influence, followed by cross-modal that further integrates information across modalities through attention mechanisms and selective enhancement. The result is a set of recovered prompts that maintain balanced representations from both modalities.

## 4 EXPERIMENT

### 4.1 SET UP

**Dataset**. Following Zhang et al. (2023), we conduct experiments on two datasets: VQA v2 (Goyal et al., 2017) and NExT-QA (Xiao et al., 2021). VQA v2 contains 1.1 million pairs of real-world images and human-written questions, while NExT-QA includes 52K annotated video-based question-answer pairs. In this paper, we consider 3 incremental settings: Question Increment (**QI**), Class Increment (**CI**), and Dual Increment (**DI**) (Zhang et al., 2023). In QI, each task introduces new question types while sharing all object classes across tasks. In CI, each task adds new object classes with shared all question types. DI combines both, in which each task has $S$ subtasks sharing the same visual classes but introducing new question types, while visual classes differ between tasks. For VQA v2, we structure the CL experiments into 8 sequential tasks for DI and CI, while 10 sequential tasks for QI. In QI, each task introduces 1 new question type. For CI, each task contains 10 unique object classes. Under DI, each task contains 5 subtasks and each subtask focuses on 2 different question types about the same 10 object classes. For NExT-QA, we structure the experiments into 7 sequential tasks. For QI, each task introduces 1 new question type. For CI, 4 tasks contain 11 object classes each, and 3 tasks contain 12 object classes each. For DI, each task contains unique question types with 5 subtasks, where each subtask addresses the same 16 object classes.

**Evaluation Metrics**. We evaluate model's performance based on two matrices: Average Performance ($A$), calculated as $A = \frac{1}{T} \sum_{t=1}^{T} \text{Acc}_{T,t}$, where $\text{Acc}_{T,t}$ means accuracy on task $t$ after training on task $T$, measures overall accuracy after the training; Inter Task Forgetting ($F_{\text{inter}}$), computed as $F_{\text{inter}} = \frac{1}{T-1} \sum_{t=1}^{T-1} \left( \max_{j \in \{1,...,t\}} \text{Acc}_{j,t} - \text{Acc}_{T,t} \right)$ quantifies the average gaps between the best performance and the current performance across tasks. In addition, for the DI setting, we introduce Inner Task Forgetting ($F_{\text{intra}}$), which is logically similar to the $F_{\text{inter}}$, and measures performance degradation across subtasks within a task.

**Implementation Details**. We construct MM-Prompt as illustrated in Fig. 2. The pretrained transformer backbone consists of 12 stacked blocks for both encoder and decoder, with each attention layer containing 12 attention heads. All experiments were conducted on a single NVIDIA RTX 4090 GPU with 24GB of memory. Following DualPrompt (Wang et al., 2022a), we further divide our question and visual prompts into General (G) and Expert (E) types that target different transformer layers. We also include a weight decay (Krogh & Hertz, 1991) on the $\mathcal{L}_{\text{inter}}$ to prevent degeneration of this alignment mechanism.

### 4.2 MAJOR RESULTS

We evaluate MM-Prompt against 9 outstanding and state-of-the-art methods, including 6 general CL methods (Wang et al., 2022a; Smith et al., 2023; Wang et al., 2022b; Khattak et al., 2023; Menabue et al., 2024) and 3 methods for CVQA (Qian et al., 2023; Zhang et al., 2023; Cai & Rostami, 2025).

Table 1: Performance comparison on VQA v2 and NExT-QA. L2P* means L2P with a single prompt pool. The best performance is highlighted in **bold**, and the second-best is underlined.

| Method | VQA v2 | | | | | | | NExT-QA | | | | | | |
| | DI | | | CI | | QI | | DI | | | CI | | QI | |
| | $A(\uparrow)$ | $F_{inter}(\downarrow)$ | $F_{intra}(\downarrow)$ | $A(\uparrow)$ | $F_{inter}(\downarrow)$ | $A(\uparrow)$ | $F_{inter}(\downarrow)$ | $A(\uparrow)$ | $F_{inter}(\downarrow)$ | $F_{intra}(\downarrow)$ | $A(\uparrow)$ | $F_{inter}(\downarrow)$ | $A(\uparrow)$ | $F_{inter}(\downarrow)$ |
|---|---|---|---|---|---|---|---|---|---|---|---|---|---|---|
| Dual Prompt (Wang et al., 2022a) | 33.601 | 2.660 | 10.574 | 36.063 | 4.449 | 14.146 | 23.518 | 16.163 | 9.591 | 9.974 | 10.693 | 1.663 | 12.393 | 11.803 |
| L2P (Wang et al., 2022b) | 31.186 | 2.112 | 12.541 | 33.706 | 4.323 | 13.720 | 23.807 | 14.181 | 9.441 | 9.263 | 9.903 | 1.635 | 10.904 | 10.081 |
| L2P* (Wang et al., 2022b) | 31.343 | 2.201 | 12.723 | 33.428 | 4.150 | 13.853 | 23.651 | 14.452 | 9.316 | 9.473 | 10.207 | 1.641 | 11.204 | 10.152 |
| CODA (Smith et al., 2023) | 35.138 | 0.902 | 10.735 | 37.102 | 4.527 | 15.438 | 22.561 | 15.115 | 9.282 | 9.361 | 11.479 | 1.556 | 22.416 | 6.960 |
| Triplet (Qian et al., 2023) | 32.826 | 1.134 | 12.244 | 37.102 | 4.527 | 14.382 | 22.697 | 18.781 | 8.674 | 9.633 | 11.362 | 1.511 | 18.719 | 9.587 |
| Maple (Khattak et al., 2023) | 35.187 | 1.054 | 11.148 | 37.450 | 4.606 | 15.371 | 22.164 | 17.877 | 8.538 | 9.394 | 11.596 | 1.641 | 16.501 | 10.588 |
| VQACL (Zhang et al., 2023) | 34.224 | 0.867 | 10.626 | 36.950 | 4.431 | 15.525 | 22.452 | 20.472 | 8.772 | 9.041 | 10.787 | 1.818 | 14.870 | 13.500 |
| Star-prompt (Menabue et al., 2024) | 34.437 | 1.784 | 10.584 | 36.516 | 4.172 | 15.104 | 22.837 | 16.695 | 10.347 | 9.218 | 11.030 | 1.742 | 15.571 | 11.750 |
| CluMo (Cai & Rostami, 2025) | 35.079 | 1.580 | 11.300 | 36.462 | 4.039 | 13.208 | 25.016 | 18.451 | 9.183 | 8.904 | 11.519 | 1.559 | 17.971 | 11.005 |
| **MM-Prompt** | **36.223** | **0.447** | **10.055** | **39.240** | **3.748** | **16.757** | **21.546** | **22.261** | **8.392** | **8.454** | **13.267** | **1.236** | **24.988** | **6.650** |

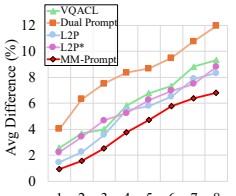

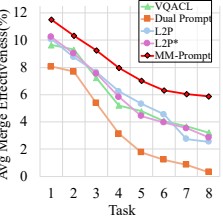

(a) Modality Diff  (b) Modality Merge

Figure 3: Comparison between existing prompt-based approaches and our MM-Prompt model.

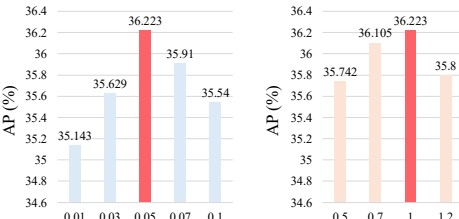

(a) Effect of different $\delta$  (b) Effect of different $\alpha$

Figure 4: Performance variation under different masking and cross alignment loss ratio.

For fair comparison, all models contain at least two types of prompts(Q and V) except L2P* and use approximately the same number of prompts with identical injection layers if possible. As shown in Table 1, we draw several important conclusions: (1) Methods adopting cross-modal prompt isolation like Dual Prompt (Wang et al., 2022a) and L2P (Wang et al., 2022b) consistently show higher forgetting rates and lower accuracy across settings, confirming that isolated prompt selection and injection lead to degradation in performance due to imbalanced representations. (2) MM-Prompt consistently outperforms all the other methods across all settings. On VQA v2, it achieves 36.223% in DI (vs. MaPLe's 35.187%), 39.240% in CI (vs. MaPLe's 37.450%), and 16.757% in QI (vs. VQACL's 15.525%). Notably, it reduces forgetting to less than half of the next best method in DI. On NExT-QA, MM-Prompt similarly outperforms in accuracy and maintains the lowest forgetting rates. While the second-best method varies across different incremental scenarios, MM-Prompt's consistent superiority highlights its robustness across diverse settings. (3) We observe striking dataset-dependent difficulty patterns: VQA v2 (static images) shows poorest performance on QI, while NExT-QA (videos) struggles most with CI, suggesting that temporal dynamics fundamentally alter how modality conflicts during continual learning. (4) Among comparison methods, MaPLe (Khattak et al., 2023) performs relatively well, which we attribute to its gradient-based visual-linguistic interactions. (5) Specialized CVQA methods like Triplet (Qian et al., 2023) and VQACL (Zhang et al., 2023) underperform despite their success in classification-based settings and with memory buffers, highlighting MM-Prompt's efficiency for maintaining balanced representations across modalities. These results collectively validate our approach's effectiveness in maintaining balanced cross-modal representations across diverse incremental learning scenarios.

## 4.3 MODALITY INTEGRATION ANALYSIS

To demonstrate that isolated prompts amplify modality imbalance, we train the models as normal and evaluate them on joint, vision-only and question-only inputs, then compute Modality Merge Effectiveness and Modality Difference, details for these settings are in Appendix A. As shown in Fig. 3, existing prompt-based methods (Wang et al., 2022a;b; Zhang et al., 2023) that adopt cross modal prompts isolation show low merge effectiveness and high modality gaps during the continual learning process. In contrast, our method maintains consistently high merge effectiveness and minimal modality disparity across new tasks, while achieving superior overall accuracy. These improvements stem from our two-component design: cross-modal prompt query prevents selection bias by enriching queries with complementary information, while cross-modal prompt recovery creates explicit pathways for modality interaction and fulfill them with cross-modal information. Together,

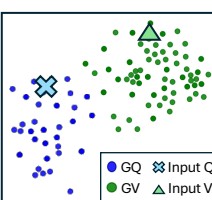 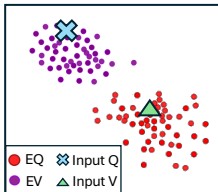 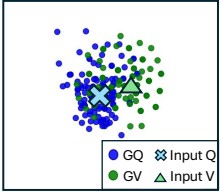 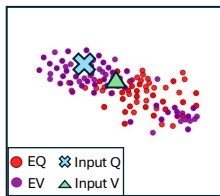

Figure 5: Comparisons of bottom-up attention visualization during inference.

(a) Dual Prompt ($A$ : 33.601, $F_{\text{inter}}$ : 2.660)  (b) MM-Prompt ($A$ : 36.223, $F_{\text{inter}}$ : 0.447)

Figure 6: Visualization of injected prompts and input feature using t-SNE.

these mechanisms counteract modality imbalance at critical points where isolation typically occurs in existing methods.

Figs. 5 and 6 provide further qualitative evidence. We visualize the attention from the model on the bounding boxes during the inference using Anderson et al. (2018). The attention maps (Fig. 5) show MM-Prompt precisely focuses on relevant image regions (correctly identifying "broccoli" and "airport"), while existing methods such as Dual Prompt produce incorrect answers with diffuse attention patterns, indicating possible occurrence of modality bias. The t-SNE visualization (Van der Maaten & Hinton, 2008) of injected prompts (Fig. 6) confirms the cross-modal merging effectiveness of MM-Prompt. As shown in Fig. 6b, MM-Prompt's prompts form more tightly assembled clusters compared to Dual Prompt in Fig. 6a, where prompts remain clearly separated by modality. This observation demonstrates how our cross-modal prompt query and recover mechanisms help prompts encode integrated information across modalities rather than remaining isolated within modality-specific subspaces. Additionally, examining the input features from the question "What green vegetable is on the plate?" in Fig. 5 reveals another key difference: in MM-Prompt, input features position at the intersection between Q and V prompt clusters, while prompts in Dual Prompt remain distinctly separated by modality. This demonstrates that MM-Prompt's injected prompts contains shared representational properties that facilitate effective cross-modal understanding and prevent modality-isolated prompts throughout sequential learning.

## 4.4 ABLATION STUDY

**Performance with Memory Buffer**. For NExT-QA dataset, which contains complex video question-answering tasks with longer sequences and fewer training data, we also evaluate models with a memory buffer of 500 samples (Table 2). As a result, MM-Prompt substantially outperforms all the other methods, given that all methods benefit from the same memory capacity. These results confirm that MM-Prompt's approach to create explicit pathways for modality interaction remain effective when supplemented with traditional continual learning techniques (Rolnick et al., 2019).

**Effect of Components**. To evaluate MM-Prompt's design, we conducted an ablation study with sequential component addition (Table 3). The baseline achieves 34.172% accuracy with 1.31 forgetting. Adding cross-modal prompt query (Eqs. (1) and (3)) improves accuracy to 34.99% and significantly reduces forgetting to 0.588, confirming the importance of cross-modal awareness during selection. The complete MM-Prompt with cross-modal

Table 3: Ablation study on VQA v2 under DI setting showing the effects of CQ: cross-modal prompt query, CR: cross-modal prompt recovery

| CQ | CR | $A$ ($\uparrow$) | $F_{\text{inter}}$ ($\downarrow$) |
|----|----|------|-------|
|    |    | 34.172 | 1.310 |
| ✓  |    | 34.990 | 0.588 |
| ✓  | ✓  | **36.223** | **0.447** |

prompt recovery (Eqs. (4) to (10)) achieves the best performance with 36.223% accuracy and 0.447 forgetting, demonstrating the effectiveness for creating explicit pathways to let prompts learn complementary information from the other modality. Together, these results reveal the complementary neatural of our two components and confirm the importance of cross-modal aware prompts.

Table 2: Performance comparison of different methods on NExT-QA dataset under DI setting with memory usage (Mem).

| Methods | Mem | $A$ ($\uparrow$) | $F_{\text{inter}}$ ($\downarrow$) |
|---|---|---|---|
| Triplet (Qian et al., 2023) | 500 | 27.996 | 3.282 |
| Dual Prompt (Wang et al., 2022a) | 500 | 27.641 | 3.765 |
| CODA (Smith et al., 2023) | 500 | 30.632 | 2.471 |
| Maple(Khattak et al., 2023) | 500 | 30.049 | 3.112 |
| VQACL (Zhang et al., 2023) | 500 | 30.753 | 2.692 |
| CluMo (Cai & Rostami, 2025) | 500 | 29.776 | 3.248 |
| **MM-Prompt** | **500** | **32.394** | **2.259** |
| MM-Prompt | 200 | 29.945 | 3.112 |
| MM-Prompt | 1000 | 34.268 | 2.427 |

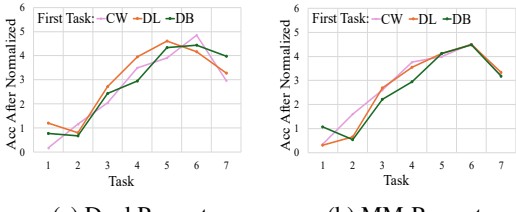

(a) Dual Prompt  (b) MM-Prompt

Figure 7: Comparison with different task orders in NExt QA under DI setting.

**Impact of Hyperparameters**. We evaluate MM-Prompt's sensitivity to two key hyperparameters in Fig. 4. For mask ratio ($\delta$), which controls the proportion of prompt tokens masked for recovery, we observed that too few masking ($\delta = 0.01$) is insufficient for prompts to effectively incorporate cross-modal information, resulting in weaker performance. Conversely, excessive masking ($\delta = 0.10$) creates too many information gaps, making results unstable and hard to reproduce. The optimal value ($\delta = 0.05$) balances these constraints, creating sufficient recovery opportunities while maintaining enough context for effective learning. Similarly, for the inter alignment loss weight ($\alpha$), we find moderate values perform best, with peak performance at $\alpha = 1.0$. Lower values provide insufficient cross-modal alignment constraints, failing to effectively counter modality isolation. Higher values ($\alpha > 1.2$) divert the model's focus from generate correct answers to alignment, degrading task performance by over-emphasizing representational similarity.

**Impact of Task Order**. Figs. 7a and 7b demonstrate MM-Prompt's stability across different learning scenarios. MM-Prompt maintains more consistent performance across different task orders, while Dual Prompt(Wang et al., 2022a) exhibits more varied performance. By maintaining balanced representations through explicit cross-modal pathways in injected prompts, MM-Prompt reduces the cascading effects of task order biases that typically compound in existing prompt methods. This order-invariant behavior strongly evidences that MM-Prompt successfully maintains balanced representations throughout the CL process.

**Computational Efficiency**. Fig. 8 shows that the inference efficiency of MM-Prompt remains competitive with existing methods. Processing 100 samples takes 0.179 seconds, which is more efficient than MAPLE (Khattak et al., 2023) (0.211s) and TRIPLET (Qian et al., 2023) (0.226s), while only slightly slower than Dual Prompt

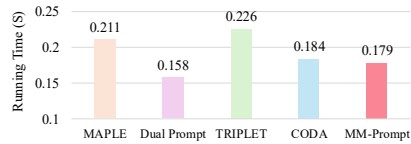

Figure 8: Inference time comparison

(Wang et al., 2022a) (0.158s). This modest computational overhead delivers substantial performance improvements and forgetting reduction compared to the Dual Prompt (Wang et al., 2022a). This demonstrates that fostering the cross-modal understanding can be achieved with reasonable computational costs, making MM-Prompt practical for real-world CL applications.

## 5 CONCLUSION

**Conclusion**. This paper presents MM-Prompt, a novel prompt-based framework for CVQA that explicitly addresses the overlooked issue of cross-modal prompt isolation. While existing methods rely on independently selected visual and textual prompts, they often worsen modality imbalance. MM-Prompt introduces a unified solution that promotes balanced and integrated cross-modal reasoning. Through two key components, cross-modal prompt query, which injects complementary modality cues into prompt selection, and cross-modal prompt recovery, which reconstructs shared representations via structured masking and hierarchical integration, MM-Prompt enables consistent fusion of multi-modal knowledge. Extensive experiments on VQA v2 and NExT-QA under various continual learning settings demonstrate that MM-Prompt achieves superior performance, enhanced modality balance, and reduced forgetting. These results confirm that addressing prompt isolation is critical for long-term, robust multi-modal learning.

**Limitations**. The current masking strategy is random, which, in a multi-modal setting, may result in the loss of important information and hinder effective training. Future work could explore adaptive and context-aware masking strategies that dynamically preserve critical cross-modal content based on different task scenarios.

ETHICS STATEMENT

This work proposes a new multi modal continual learning framework and does not involve human subjects, sensitive data, or foreseeable harmful applications. Therefore, we believe it does not raise specific ethical concerns.

REPRODUCIBILITY STATEMENT

We provide all code, training scripts, and data preprocessing steps in the supplementary materials to facilitate replication of our results.

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

# APPENDIX

## A DATASETS USED IN OUR WORK

### A.1 DATASET PARTITIONING

For VQA v2 (Goyal et al., 2017), we organize the continual learning experiments into 8 sequential tasks for both Class Increment (CI) and Dual Increment (DI) settings. In the CI setting, each task introduces 10 unique object classes. In the DI setting, each task consists of 5 subtasks, each focusing on 2 distinct question types about the same 10 object classes. For the Question Increment (QI) setting, we define 10 sequential tasks, each introducing 1 new question type.

For NExT-QA (Xiao et al., 2021), we structure the experiments into 7 sequential tasks due to the dataset's scale. In the QI setting, each task introduces 1 new question type. In the CI setting, 4 tasks contain 11 object classes each, while the remaining 3 tasks contain 12 object classes each. In the DI setting, each task consists of 5 subtasks with unique question types, each subtask cover 16 different object classes.

Table 4: Object class increment in VQA v2 for CI and DI.

| Task | Objects |
|---|---|
| Group 1 | truck, couch, bowl, chair, scissors, sandwich, orange, knife, dining table, potted plant |
| Group 2 | teddy bear, microwave, skateboard, bottle, hot dog, book, apple, refrigerator, tennis racket, oven |
| Group 3 | laptop, person, car, banana, snowboard, bed, umbrella, surfboard, motorcycle, sink |
| Group 4 | tv, dog, baseball bat, cup, parking meter, sheep, spoon, cake, broccoli, toilet |
| Group 5 | kite, fork, bus, train, cell phone, pizza, keyboard, sports ball, cow, giraffe |
| Group 6 | baseball glove, bear, wine glass, traffic light, horse, mouse, stop sign, zebra, handbag, skis |
| Group 7 | fire hydrant, toaster, cat, bench, remote, clock, hair drier, bird, suitcase, toothbrush |
| Group 8 | carrot, backpack, tie, elephant, frisbee, bicycle, donut, boat, airplane, vase |

### A.2 PARTITIONING DETAILS AND GROUP DEFINITIONS

For VQA v2, we structure the continual learning experiments into 8 sequential tasks for DI and CI, while QI has 10 sequential tasks due to the total number of question types. Tables 4, 6, and 7 provide details on the distribution of object classes and question types across these tasks. In the DI setting, tasks follow the class distribution outlined in Table 4, with each task further divided into 5 subtasks according to the question type distribution in Table 6. For the CI setting, we directly implement the class grouping specified in Table 4 while maintaining all question types across each task. In the QI setting, we utilize the question grouping from Table 7, while keeping object classes consistent across all tasks.

For NExT-QA (Xiao et al., 2021), we structure the experiments into 7 sequential tasks following the approach of Zhang et al. (2023). In the DI setting, we implement the hierarchical structure defined in Zhang et al. (2023), with question increments ordered according to Table 8. For the QI setting, we utilize the question type ordering specified in Table 8 while maintaining consistent visual classes across all tasks. In the CI setting, we employ the object class groups outlined in Table 5 while preserving all question types throughout the tasks.

Table 5: Object classes increment in NExT-QA for CI.

| Task | Objects |
|------|---------|
| Group 1 | bicycle, camel, bat, microwave, snake, sofa, traffic light, hamster/rat, chicken, oven, stop sign |
| Group 2 | crab, camera, lion, ball/sports ball, crocodile, screen/monitor, baby walker, cat, squirrel, frisbee, cattle/cow |
| Group 3 | piano, watercraft, kangaroo, train, fruits, pig, suitcase, bear, tiger, bench, elephant |
| Group 4 | ski, stingray, antelope, toy, child, duck, guitar, dish, fish, cake, turtle, leopard |
| Group 5 | penguin, faucet, car, bottle, bus/truck, aircraft, baby, bread, baby seat, cellphone, sink, rabbit |
| Group 6 | vegetables, skateboard, bird, toilet, racket, sheep/goat, adult, scooter, electric fan, stool, motorcycle |
| Group 7 | horse, snowboard, surfboard, handbag, laptop, panda, table, cup, backpack, chair, dog, refrigerator |

## B   DETAILED COMPONENT ANALYSIS

### B.1   MODALITY MERGE EFFECTIVENESS AND MODALITY DIFFERENCE

In Section 4.3, we use Modality Merge Effectiveness and Modality Difference to demonstrate that isolated prompts amplify modality imbalance. To get these two matrices, we train models under the standard setting and evaluate them using three input configurations, joint, vision-only, and question-only. Remind that $\mathbf{F}^Q$ and $\mathbf{F}^V$ denote the question and vision features, $\mathbf{p}_{final}^Q$ and $\mathbf{p}_{final}^V$ represents the final question and vision prompts that will be injected to the encoder.

In the joint setting, both modality features $\mathbf{F}^Q$ and $\mathbf{F}^V$, as well as the final prompts $\mathbf{p}_{final}^Q$ and $\mathbf{p}_{final}^V$, are available to the model. The vision-only setting provides only $\mathbf{F}^V$ and $\mathbf{p}_{final}^V$, while the question-only setting provides only $\mathbf{F}^Q$ and $\mathbf{p}_{final}^Q$. To quantify modality interaction and imbalance, we compute:

$$\text{Modality Merge Effectiveness} = \text{Acc}^{\text{joint}} - \max(\text{Acc}^{\text{V-Only}}, \text{Acc}^{\text{Q-Only}}). \quad (11)$$

This metric quantifies the additional information gained through cross-modal integration. Higher values indicate the model effectively leverages complementary signals from both modalities beyond what either modality provides alone.

$$\text{Modality Difference} = \max(\text{Acc}^{\text{V-Only}}, \text{Acc}^{\text{Q-Only}}) - \min(\text{Acc}^{\text{V-Only}}, \text{Acc}^{\text{Q-Only}}). \quad (12)$$

This metric measures the performance gap between modalities, where higher values indicate greater reliance on the dominant modality, while lower values reflect more balanced representations across the two modalities.

Ideally, a model should maximize Merge Effectiveness and minimize Modality Difference. MM-Prompt achieves consistently higher Modality Merge Effectiveness and lower Modality Difference across sequential tasks compared to all baseline methods. This quantitative evidence confirms that our approach not only maintains more balanced representations of both modalities but also more effectively integrates complementary information across them. These results further validate our design choices: Cross-Modal Prompt Query ensures prompts inherently carry the cross-modal information, while Cross-Modal Prompt Recovery reinforces integration through explicit information exchange pathways, together creating a robust defense against the modality isolation that typically accumulates in CL scenarios.

Table 6: Question types increment in VQA v2 for DI.

| Task | Group 1 | Group 2 | Group 3 | Group 4 | Group 5 |
|------|---------|---------|---------|---------|---------|
| **Question Types** | Recognition Location | Judge Commonsense | Count Action | Color Type | Subcategory Causal |

Table 7: Question types increment in VQA v2 for QI.

| Task | G1 | G2 | G3 | G4 | G5 |
|------|----|----|----|----|----|
| Question Type | Recognition | Location | Judge | Commonsense | Count |
| Task | G6 | G7 | G8 | G9 | G10 |
| Question Type | Action | Color | Type | Subcategory | Causal |

Table 8: Question types increment in NExT-QA for QI and DI.

| Task | G1 | G2 | G3 | G4 | G5 | G6 | G7 |
|------|----|----|----|----|----|----|----|
| Question Type | CW | TN | TC | DL | DB | DC | DO |

## B.2 INSERTION LAYERS AND NUMBER OF PROMPTS

Table 9: Effect of different number of prompts in the order of QG, QE, VG, VE.

| Number of Prompts | $A (\uparrow)$ | $F_{\text{inter}}(\downarrow)$ |
|-------------------|----------------|--------------------------------|
| 10,80,10,80 | 35.79 | 0.914 |
| 50,50,50,50 | 35.168 | 1.268 |
| 40,60,80,120 | **36.223** | **0.447** |
| 60,80,100,140 | 34.851 | 0.985 |

Following Wang et al. (2022a), we insert General prompts at lower layers {1,2} and Expert prompts at intermediate layers {3,4,5}. This configuration follows the hierarchical processing pattern of the transformer, where lower layers handle more abstract, transferable representations, while intermediate layers process more specific, fine-grained patterns.

As shown in Table 9, we set the number of prompts to be QG=40, QV=60, EG=80, EV=120, allocating more capacity to visual prompts and expert prompts, as there are more object classes than question types in the dataset. The larger number of Expert prompts enables more specialized handling of diverse visual concepts and complex reasoning patterns, improving the model's ability to adapt to new tasks. Notably, simply increasing prompt counts (60,80,100,140) decreases performance, suggesting that excessive prompt quantities can lead to lower representation quality and slower convergence.

## B.3 IDENTICAL MASKING

As shown in Figure 9, we use formula $\frac{\text{Identical - Separate}}{\text{Identical}}$ to calculate the relative difference between identical masking and separate masking, the results reveal consistently positive values for both modality merge effectiveness and modality difference across all tasks. We attribute this improvement to the more aligned reconstruction objectives created by Identical masking through masking the same regions in both modalities. By doing so, it establishes shared targets that incentivize leveraging complementary cross-modal information, leading to better cross-modal awareness than separate masks. This mechanism effectively embeds cross-modal awareness into the recovered prompts, leading to the consistent performance improvements observed across all continual learning tasks compare with separate masking for 5% masking ratio.

## B.4 INTRA AND INTER MODAL RECOVERY ANALYSIS

Table 10 presents an ablation study examining the contribution of each recovery step to MM-Prompt's performance. The baseline model with only cross-modal query achieves 34.990% AP with 0.588 forgetting. Adding only intra-modal recovery marginally improves accuracy to 35.150% but substantially worsens forgetting to 1.140, nearly doubling the forgetting rate. This reveals a key insight that intra-modal recovery alone may intensify the modality imbalance problem by strengthening modality-specific patterns without proper cross-modal integration. On the other hand, without the intra-modal phase to first preserve and emphasize essential modality-specific characteristics, inter-modal recovery causes premature mixing where important uni-modal patterns are lost or dom-

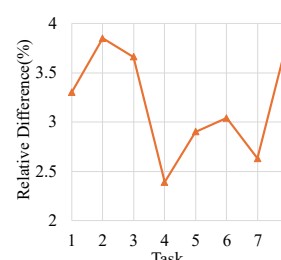

(a) Relative difference on modality difference

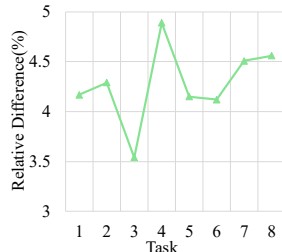

(b) Relative difference on modality merge effectiveness

Figure 9: Relative difference between identical masking positions and separate masking positions

Table 10: Performance variation for different recovery components

| Intra | Inter | Gate | $A$ ($\uparrow$) | $F_{\text{inter}}$($\downarrow$) |
|---|---|---|---|---|
| | | | 34.990 | 0.588 |
| ✓ | | | 35.168 | 1.268 |
| | | ✓ | 35.318 | 1.071 |
| ✓ | ✓ | | 35.916 | 0.526 |
| ✓ | ✓ | ✓ | **36.223** | **0.447** |

inated by the stronger modality before they can be properly established. When both phases are applied, we can see significant improvements in both metrics (35.916% AP, 0.447 forgetting). This demonstrates that balanced cross-modal representations require both the preservation of modality-specific characteristics provided by intra-modal recovery and explicit pathways for cross-modal integration provided by inter-modal recovery, working together to overcome the isolation problem that undermines existing approaches. Finally, incorporating the gating mechanism further improves performance to 36.223% AP with 0.447 forgetting. The gating selectively applies refinement only where cross-modal evidence supports it, enabling targeted enhancement.

Table 11: Performance variation for different alignment loss

| $\mathcal{L}_{\text{intra}}$ | $\mathcal{L}_{\text{inter}}$ | $A$ ($\uparrow$) | $F_{\text{inter}}$($\downarrow$) |
|---|---|---|---|
| | | 34.260 | 1.871 |
| ✓ | | 34.743 | 1.531 |
| | ✓ | 35.059 | 1.204 |
| ✓ | ✓ | **36.223** | **0.447** |

Table 11 demonstrates the importance of both $\mathcal{L}_{\text{intra}}$ and $\mathcal{L}_{\text{inter}}$ for effective prompt recovery. Without any alignment losses, the model achieves only 34.260% accuracy with high forgetting (1.871), as the recovery process lacks proper guidance and may not be able to reconstruct any meaningful information. The $\mathcal{L}_{\text{intra}}$ ensures accurate reconstruction of modality-specific patterns, preserving essential characteristics of each modality. Without this foundation, the model would struggle to maintain the fundamental representation structure needed for effective reasoning within each modality. Meanwhile, the $\mathcal{L}_{\text{inter}}$ explicitly guides cross-modal interaction, preventing one modality from dominating the recovery process and ensuring balanced integration. Only when both losses work together, the model achieve the optimal performance by simultaneously preserving modality-specific characteristics while enforcing balanced cross-modal integration.

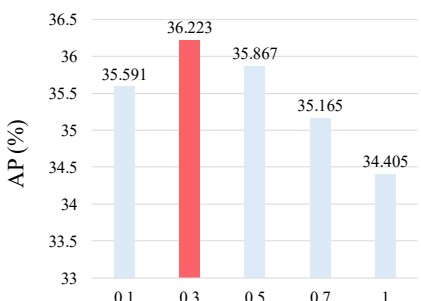

Figure 10: Effect of different $\mathcal{L}_{\text{intra}}$ ratio.

Fig. 10 shows the effect of different $\mathcal{L}_{\text{intra}}$ ratios. Similar to the ratio for $\mathcal{L}_{\text{inter}}$, the moderate value still performs the best, with peak performance at $\alpha = 0.3$. Lower values provide insufficient reconstruction guidance, fail to reconstruct the original representations. Higher values shift the model's focus away from producing correct answers to enforcing alignment, which harms task performance by placing excessive emphasis on representational similarity.

### B.5 COMPARISONS WITH OTHER FUSION STRATEGIES

Table 12: Comparisons of different multi-modal fusion strategies during the query stage.

| Strategy | $A$ ($\uparrow$) | $F_{\text{inter}}$($\downarrow$) |
|---|---|---|
| Plus | 34.275 | 1.731 |
| Mean Pooling | 34.398 | 2.015 |
| Hadamard Product | 34.702 | 1.142 |
| Cross Attention | 34.891 | 1.202 |
| Ours' Query | **36.223** | **0.447** |

Table 13: Performance variation under different strategies of modality interaction before prompt injection.

| Strategy | $A$ ($\uparrow$) | $F_{\text{inter}}$($\downarrow$) |
|---|---|---|
| Plus | 33.850 | 1.335 |
| Mean Pooling | 34.170 | 1.405 |
| Hadamard Product | 34.212 | 1.274 |
| Cross Attention | 34.628 | 1.428 |
| Our Recovery | **36.223** | **0.447** |

Tables 12 and 13 present comprehensive analysis of different modality interaction strategies at the query stage and before the injection stage. In Table 12, we compare our cross-modal prompt query against simpler integration methods. Naive strategies like "Plus", "Mean Pooling" and "Hadamard Product" directly combining features from both modalities achieve low accuracy with high forgetting, as a simple combination provides no mechanism to balance the influence of the dominant modality, which can overwhelm the joint representation. The standard "Cross Attention" approach improves performance moderately by allowing selective feature integration, but still permits the dominant modality's features to disproportionately influence attention weights and outputs. In contrast, our cross-modal prompt query approach with controlled residual connections and learnable modulation weights outperforms all the alternatives by maintaining essential modality-specific characteristics while enabling balanced cross-modal influence.

Table 13 further demonstrates that our structured approach for prompt recovery is equally crucial. Cross-modal prompt recovery creates explicit pathways for cross-modal information integration. Our recovery method first establishes modality-specific foundations before progressively integrating cross-modal information through controlled pathways. This structured approach preserves essential characteristics of each modality while enabling targeted information exchange. Unlike the naive combination methods that allow dominant signals to overwhelm the integration process, our method creates explicit, balanced pathways for cross-modal information flow.

Table 14: Performance comparison over 10 times long sequence (10 times longer sequence - original).

| Method | After Task | Acc on that Task | A ($\uparrow$) | $F_{inter}$ ($\downarrow$) |
|---|---|---|---|---|
| Dual Prompt | Task 1 | +1.26 | +1.26 | 0.00 |
| Dual Prompt | Task 2 | +1.54 | +0.49 | +1.91 |
| Dual Prompt | Task 3 | +1.89 | +0.04 | +2.28 |
| Ours | Task 1 | +2.57 | +2.57 | 0.00 |
| Ours | Task 2 | +2.77 | +2.01 | +1.32 |
| Ours | Task 3 | +3.22 | +1.73 | +1.68 |

Notably, the "Plus" operation yields the lowest performance in both stages, suggesting that simply adding the prompts together can significantly diminishes the complementary information provided by the vision modality. These results confirm that both components of cross-modal prompt query and cross modal prompt recovery complementarily creating prompts with effective cross-modal awareness, thus achieve the best performance.

## B.6 ON SCALING WITH EXTREMELY LONG CONTINUAL LEARNING SEQUENCES

To evaluate our method's robustness on longer sequences, we conducted experiments using 10% of the original batch size on VQA v2 CI setting to simulate 10 times longer continual sequence. All The results are calculated by 10 times long sequence minus standard. Results are shown in the Table 14, we attribute the performance change in this setting to a more challenging learning dynamic. With smaller batch size, the model may optimize too rapidly toward each task's specific set of image classes, exacerbating conflicts between different tasks. Despite these challenges, our model demonstrates superior performance. It achieves substantially higher improvements in both per-task and average accuracy, while its increase in inter-task forgetting is less than half that of the existing method. This confirms our cross-modal mechanisms effectively mitigate degradation by preserving balanced modality representations, while isolated prompt methods suffer from compounding modality bias.

## B.7 MORE STATISTICS

**Statistical Reliability** To validate the robustness of MM-Prompt, we conducted experiments with three different random seeds on both VQA v2 DI and NExT-QA DI settings, reporting mean and standard deviation in Table 15. MM-Prompt achieves the highest average performance across all metrics while maintaining the lowest or near-lowest standard deviation. These consistent results across multiple runs demonstrate that fostering cross-modal interaction through explicit pathways is not only effective but also stable. The low variance indicates that cross-modal prompt query and cross-modal prompt recovery can always guide the model toward balanced multimodal representations regardless of initialization conditions. This stability is important for continual learning scenarios where performance consistency ensures reliable deployment across diverse applications.

**Robustness Across Different Backbones** To further evaluate the generalizability of our approach, we test MM-Prompt with a different pretrained backbone. Table 16 presents results using Flan-T5 (Chung et al., 2024) and Bart (Lewis et al., 2020) as the backbone model. In both settings, our method achieves achieves the highest accuracy and lowest forgetting. This consistent superior performance confirms that our cross-modal prompt mechanisms are robust across different pretrained backbones and not dependent on specific pretraining strategies, suggesting the broader applicability of our approach.

**First-task Accuracy Across Sequential Learning** Table 17 presents accuracy for the Task 1 after training on each tasks. MM-Prompt consistently maintains the highest accuracy on Task 1 throughout the entire learning sequence and achieved the lowest forgetting in the end. We attribute this superior retention to our cross-modal prompt query and recovery mechanisms, which effectively embed the complementary information in to the prompt, thereby fostering cross modal interaction.

**Component-Level Forgetting** To investigate if our learned attention modules suffer from forgetting, we conducted a new analysis to measure their attention drift on the VQA v2 DI setting. We define drift as the angular calculated by Attention_drift (task T) = $\frac{2 * \cos^{-1}\left(\frac{\mathbf{a}_t \cdot \mathbf{a}_1}{\|\mathbf{a}_t\| \|\mathbf{a}_1\|}\right)}{\pi}$. Results shown in

the Table 18 reveal two key findings:(1) Intra-modal attention shows lower drift, providing a stable modality-specific foundation. (2) Inter-modal attention show higher drift, as they aim to adapt to capture different cross-modal relationships. This combination of a stable intra-modal foundation and an adaptive inter-modal integration demonstrates a reasonable and beneficial feature of our design. This drift allows the model to adapt to new tasks while preserving foundational knowledge, ultimately leading to the state-of-the-art performance.

**Computational Efficiency** We also analyze the theoretical computational complexity by measuring Giga Floating Point Operations (GFLOPs). MM-Prompt requires 19.824 GFLOPs, which is only higher than the simplest method Dual Prompt (Wang et al., 2022a), while been lower than the other methods. Notably, the GFLOPs ranking aligns with the empirical running time measurements (Fig.8 in the main paper), further confirms that achieving balanced multimodal representations through cross-modal mechanisms does not require significant computational trade-offs, making MM-Prompt suitable for practical applications.

Table 15: Performance statistics for different models, with error bars denoting mean ± standard deviation across three trials using distinct random seeds. Superscript symbols indicate a representative pair of statistically overlapping results per metric. $^{\dagger}$, $^{\ddagger}$, $^{\S}$ and $^{\P}$ denote statistically overlap

| Methods | VQA-v2 DI | | NExT QA DI | |
|---|---|---|---|---|
| | $A$ ($\uparrow$) | $F_{\text{inter}}$ ($\downarrow$) | $A$ ($\uparrow$) | $F_{\text{inter}}$ ($\downarrow$) |
| Dual Prompt | $33.748 \pm 0.213$ | $2.630 \pm 0.248$ | $16.224 \pm 0.211^{\S}$ | $9.537 \pm 0.190$ |
| L2P | $31.481 \pm 0.255$ | $2.227 \pm 0.230$ | $14.204 \pm 0.256$ | $9.480 \pm 0.195$ |
| CODA | $35.070 \pm 0.228^{\dagger}$ | $1.026 \pm 0.211^{\ddagger}$ | $15.202 \pm 0.227$ | $9.281 \pm 0.239$ |
| Triplet | $32.602 \pm 0.204$ | $1.211 \pm 0.227$ | $18.903 \pm 0.189$ | $8.716 \pm 0.218^{\P}$ |
| Maple | $\underline{35.086} \pm 0.177^{\dagger}$ | $1.170 \pm 0.241$ | $17.636 \pm 0.250$ | $\underline{8.648} \pm 0.248$ |
| VQACL | $34.243 \pm 0.239$ | $\underline{1.008} \pm 0.217^{\ddagger}$ | $\underline{20.496} \pm 0.231$ | $8.734 \pm 0.205$ |
| CluMo | $35.037 \pm 0.270$ | $1.739 \pm 0.252$ | $18.412 \pm 0.226$ | $9.205 \pm 0.199^{\P}$ |
| Star-prompt | $34.449 \pm 0.241$ | $1.776 \pm 0.263$ | $16.787 \pm 0.237^{\S}$ | $10.315 \pm 0.232$ |
| **Ours** | $\mathbf{36.296} \pm 0.162$ | $\mathbf{0.571} \pm 0.195$ | $\mathbf{22.535} \pm 0.205$ | $\mathbf{8.442} \pm 0.162$ |

Table 16: Comparison of different backbones on VQA-v2 DI.

(a) BART-based Methods

| Method | ($\uparrow$) | ($\downarrow$) |
|---|---|---|
| Dual Prompt | 35.012 | 3.406 |
| L2P | 33.527 | 3.274 |
| CODA | 36.157 | 1.986 |
| MaPLe | 36.454 | 2.057 |
| CluMo | 35.934 | 2.377 |
| **Ours** | **38.231** | **1.251** |

(b) Flan-T5-based Methods

| Method | $A$ ($\uparrow$) | $F_{\text{inter}}$ ($\downarrow$) |
|---|---|---|
| Dual Prompt | 34.955 | 1.750 |
| L2P | 33.100 | 1.855 |
| CODA | 35.966 | $\underline{1.653}$ |
| MaPLe | $\underline{36.179}$ | 2.059 |
| CluMo | 35.355 | 1.797 |
| **Ours** | **37.572** | **1.577** |

Table 17: Task 1 performance across sequential learning.

| Method | (After) T1 | T2 | T3 | T4 | T5 | T6 | T7 | T8 | $F_{\text{inter}}$ |
|---|---|---|---|---|---|---|---|---|---|
| Dual Prompt | 32.68 | 28.77 | 24.18 | 21.45 | 16.87 | 13.93 | 11.64 | 8.93 | 23.75 |
| L2P | 32.45 | 28.01 | 23.79 | 20.82 | 15.92 | 12.61 | 11.07 | 8.17 | 24.28 |
| CODA | 34.02 | 31.47 | 28.06 | 23.82 | 19.31 | 15.50 | 12.82 | 10.56 | 23.46 |
| Triplet | 32.15 | 28.84 | 25.63 | 22.08 | 17.49 | 13.74 | 11.57 | 9.12 | 23.03 |
| MaPLe | 33.87 | 32.29 | 27.74 | 23.13 | 19.25 | 15.61 | 12.47 | 11.06 | 22.81 |
| VQACL | 33.94 | 31.38 | 27.85 | 22.76 | 18.71 | 15.45 | 13.39 | 11.28 | 22.66 |
| CluMo | 33.66 | 28.28 | 24.06 | 21.48 | 16.33 | 13.61 | 11.29 | 8.39 | 25.27 |
| Star-Prompt | 33.44 | 30.79 | 27.53 | 21.68 | 18.90 | 14.87 | 12.53 | 10.19 | 23.25 |
| **Ours** | **35.13** | **32.63** | **29.45** | **25.29** | **20.70** | **17.26** | **14.59** | **12.94** | **22.19** |

Table 18: Attention drift on different components.

| After Training On | $A_{query}$ | $A_{intra}$ | $A_{inter}$ | Average |
|---|---|---|---|---|
| Task 2 | 1.04 | 0.98 | 1.86 | 1.29 |
| Task 3 | 3.08 | 1.67 | 3.30 | 2.68 |
| Task 4 | 3.33 | 1.74 | 3.52 | 2.86 |
| Task 5 | 3.59 | 1.76 | 3.58 | 2.98 |
| Task 6 | 3.85 | 1.81 | 3.70 | 3.12 |
| Task 7 | 4.05 | 1.85 | 3.87 | 3.26 |
| Task 8 | 5.32 | 3.30 | 6.88 | **5.17** |

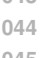

Figure 11: GFLOPs for different methods

## C  MORE VISUALIZATION

### C.1  HEATMAP COMPARISON

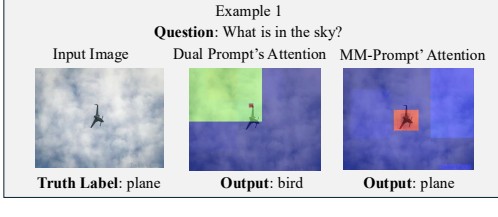

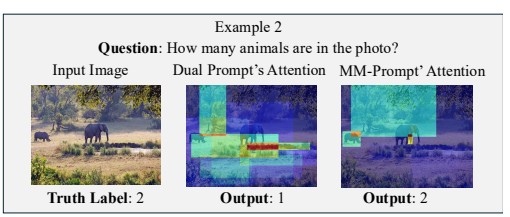

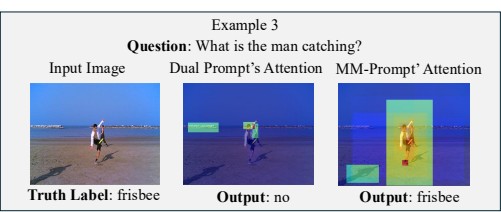

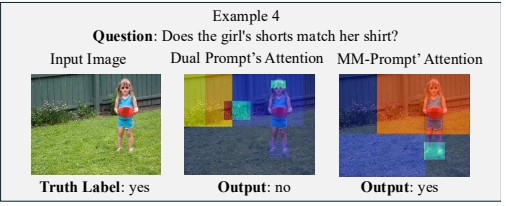

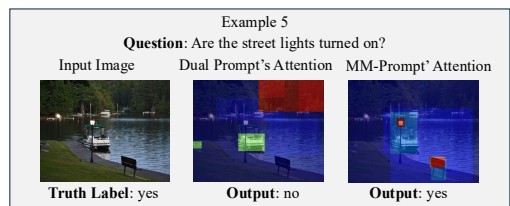

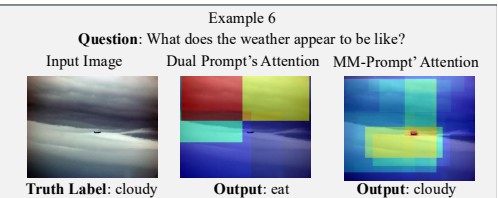

Figure 12: Bottom up attention visualization(Anderson et al., 2018) during inference on Dual Prompt (Wang et al., 2022a) and MM-Prompt.

As illustrated in Fig. 12, MM-Prompt consistently directs attention to relevant regions, while Dual Prompt often diffuses or drifts focus, leading to incorrect predictions. In Example 1, MM-Prompt focus on the aircraft in the sky and generate the correct answer "plane" while Dual Prompt attends to an edge of the wing and misclassifies it as "bird" A similar pattern holds in Example 2, MM-Prompt highlights both animals and counts "2" whereas Dual Prompt spreads attention across the pasture and under-counts. In Example 3, where fine-grained localization is required , MM-Prompt put the

attention on the frisbee in the man's hand and answers correctly, contrasting with Dual Prompt's off-target focus that yields "no". These patterns persist across other tasks involving abstract reasoning and lighting assessment. These improvements arise from the design of MM-Prompt, where cross-modal prompt query enriches queries with complementary cues and cross-modal prompt recovery established explicit pathways for modality interaction. Together, these mechanisms create prompts with cross-modal awareness, thus counteract modality imbalance and improve performance.

## C.2  FAILURE CASE

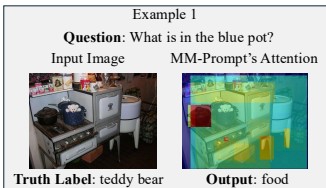 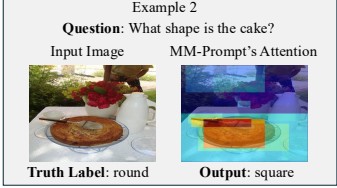 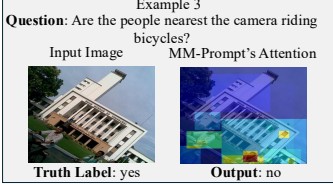

Figure 13: Failure cases for MM-Prompt

Despite MM-Prompt's overall strong performance, we observe several failure cases that reveal persistent challenges in cross-modal reasoning (Fig. 13). In Example 1, for the question "What is in the blue pot?", MM-Prompt incorrectly attends to the black pot rather than the deep blue one, indicating difficulty distinguishing objects with similar colors. Example 2 highlights limitations in shape reasoning: the model misidentifies a round cake as "square," likely due to the bounding-box-based visual features, which inadequately capture circular shapes. This suggests that when initial features are impoverished, our recovery mechanism may be insufficient. Example 3 illustrates shortcomings in spatial reasoning for the question "Are the people nearest the camera riding bicycles?". Here, attention shifts to the wrong individuals, pointing to difficulties in multi-step reasoning and hinting at potential re-emergence of modality dominance.

These failure cases point to specific areas for future improvement. (1) more adaptive masking strategies that align better with the case-specific settings, (2) more advanced visual feature representations beyond bounding boxes to better capture shape information. These enhancements would strengthen MM-Prompt's ability to maintain balanced cross-modal representations while addressing specific reasoning challenges.

## USE OF LLM

No LLMs were used in this work.

