# OpenReview forum: "MM-Prompt: Cross-Modal Prompt Tuning for Continual Visual Question Answering"
_ICLR.cc/2026/Conference — ICLR 2026 Conference Withdrawn Submission_

### Official Review · Reviewer_w2V6 · 2025-10-24

**Soundness:** 3
**Presentation:** 2
**Contribution:** 2
**Rating:** 4
**Confidence:** 3

**Summary:**

This paper focuses on continual VQA and points out that existing prompt-based methods treat visual and textual prompts separately, which leads to increasing modality imbalance and forgetting. The authors propose MM-Prompt, a framework that introduces cross-modal prompt query and recovery to fuse complementary information between modalities during prompt selection and reconstruction. Experiments on VQA v2 and NExT-QA show improved accuracy and better knowledge retention over prior approaches.

**Strengths:**

- The overall structure is clear, and the paper presents the modality imbalance problem in a straightforward way, making it easy to understand why existing methods may fail.

- The experimental evaluation is extensive, including multiple continual learning settings (e.g., task order and memory variations) and comparisons with many prior methods, which makes the results convincing.

- The authors also provide a discussion of the method’s limitations, showing awareness of the remaining challenges and giving a balanced view of the work.

**Weaknesses:**

- The work addresses a relevant problem and the proposed design is effective, but the innovation lies mainly in architectural refinements within the prompt-tuning pipeline rather than a fundamental methodological or theoretical advancement.

- All cross-modal interaction happens at the prompt level while the multimodal backbone remains pre-trained. It is unclear whether MM-Prompt improves deep multimodal reasoning or mainly adjusts the injected prompt signals. Can authors provide some additional evidence that would strengthen the claim that the method truly mitigates modality imbalance?

- What is the meaning of the MM prompt? Multimodal prompt? This shortened form is unclear.

- Some typos
  - No period at the end of lines 79 and 82.
  - In Figure 11 and Figure 12, some labels and formats are inconsistent, e.g., “Dual prompt” and “frisbee”.

**Questions:**

See weaknesses please

---

### Official Review · Reviewer_BgVP · 2025-10-27

**Soundness:** 3
**Presentation:** 2
**Contribution:** 2
**Rating:** 2
**Confidence:** 3

**Summary:**

The paper introduces a method for continual VQA  (MM-Prompt). The method integrates cross-modal prompt query and recovery to better exploit the multi-modal nature of VQA. Results on standard datasets are good.

**Strengths:**

- the proposed cross-modal interactions in the prompt selection makes sense and should improve over isolated prompts.
- ablations of the method show importance of proposed new modules
- Method obtains superior results on two commonly used datasets.

**Weaknesses:**

- Many of the cited prompting methods have not been designed for VQA. VQA having both text and visual data as input is fundamentally different from standard classification. Indeed Fig 1a does not make that much sense for VQA (also not so many papers claim it does). The paper should better explain its difference with other multi-modal prompting methods (like Khartak). Is this not also the reason for the very bad attention maps in Figure 5 ?

- joint training results are missing as upperbound. The results with replay show that there is significant forgetting (1000 exemplars) obtains 34 whereas exemplar free obtains 22 on accuracy (do I see this correct ? ) This is really huge, making me doubt much of the current setup.

- I am not sure what the study in Figure 3 proves (even though this is used as an important motivation in the introduction). The proposed method clearly encourages modalities to encode information which is already encoded by the other modality (due to masking). Figure 3 indeed shows that this redundancy exists. For me it is unclear that that is necessarily a good thing: the results in Table 3 show the proposed method indeeds improves results, but from Fig.3 we cannot conclude anything.

- Table 17 (dataset information missing in caption) shows that the method has a 1-2% head-start with respect to most methods on T1. This could mean that the proposed method is just a better VQA method, but with respect to continual VQA it is not adding anything (since the gain remains the small over tasks). The forgetting is really catastrophic.

minor
- The improvement over state-the-art is small, especially that many works are from 2023.
- would be nice to somehow see in Table 1 which methods have been originally made for VQA (because several have not)

**Questions:**

Please address the weaknesses.

I would like to see joint training results. Also Table 17 results for the other dataset would be nice. It would be good to see some analysis why there is such a large amount of catastrophic forgetting. Table 17 shows a drop from 35 to 12 for task 1 performance, this means that there are serious problems in the system and other anti-forgetting mechanisms should be introduced.

---

### Official Review · Reviewer_Cmuf · 2025-10-31

**Soundness:** 2
**Presentation:** 1
**Contribution:** 1
**Rating:** 2
**Confidence:** 3

**Summary:**

The paper introduces a novel two-stage prompt selection paradigm, MMPrompt, for injecting multimodal prompts in pretrained models. The two-stage model architecture shown for encoder-decode style models, consisting of a cross-modal prompt query module and cross modal recovery module to get modality-balanced prompts, resolves the issue of dominant modality bias for continual visual question answering tasks. The authors provide the results of different benchmarks to validate their approach

**Strengths:**

1. The approach to explicitly mitigate the dominant modality bias in pre-trained models for continual VQA is novel
2. The authors compare their approach with a variety of previous works, which validates the strength of their proposed approach
3. The authors follow an extensive ablation study validating the choice of each component used in MMPrompt.

**Weaknesses:**

1. Line 126-127 - “This reinforces alignment with modality-specific feature distributions, amplifies the dominant modality bias, and hinders the integration of complementary information.” - The authors provide no experiment showing that their pretrained model has a dominant modality bias.
2. The motivation behind developing a cross-modal prompt selection strategy is not clear. Many state-of-the-art large multimodal models, such as Qwen2-VL and LLaVA-OneVision, simply extract vision features from a pre-trained encoder and concatenate them with text tokens to achieve good performance. A comparison with these models is needed to understand this utility better.
3) The authors discuss prompt tuning but do not show any comparison with it for MMPrompt. Also, other conventional finetuning approaches like prefix tuning and LoRA can be applied for continual VQA tasks, and how does MMPrompt compare with it.
4) What is meant by identical binary masks to visual and text modality, as they are structurally different? Perhaps some qualitative examples showing how masking is performed would be helpful to better understand.
5) Figure 6a shows a gap in the injected prompts for different modalities. Why is there such a gap in pretrained models, and does this relate to the previous works on modality gap [1,2]
6) The authors only evaluate their approach on a fixed model size and on enc-dec models. What about other sizes and decoder-only architectures like Qwen-VL?
7) What is the need for a prompt pool, and how many prompts are needed to select a good weighted average for cross-modal prompt selection?

[1] Mind the Gap: Understanding the Modality Gap in Multi-modal Contrastive Representation Learning

[2] Connect, Collapse, Corrupt: Learning Cross-Modal Tasks with Uni-Modal Data

**Questions:**

1. Please show experiments showing the dominant modality bias in the pretrained model used.
2. Comparison with other decoder-only models like Qwen and LLaVA, which are widely used for VQA tasks, is required to better understand the motivation.
3. Comparison with other finetuning approaches is required to understand why prompt selection is required instead of prompt/adapter tuning.
4. Please provide some qualitative examples showing how masking is applied.
5. Please provide some explanation on why there is a modality gap in the injected prompts
6. Comparison with different sizes of models to show that MMPrompt generalizes well across the choice of the number of parameters.
7. Please explain what is the need for a prompt pool and some ablations on how large the pool should be to get good performance for MMPrompt

---

### Official Review · Reviewer_CPTC · 2025-11-04

**Soundness:** 4
**Presentation:** 3
**Contribution:** 3
**Rating:** 6
**Confidence:** 3

**Summary:**

To solve the cross-modal prompt isolation problem in Continual Visual Question Answering tasks, this work proposes MM-Prompt that includes a cross-modal prompt query module and a cross-modal prompt recovery module to help multimodal interaction.

**Strengths:**

1. This work proposed a plausible solution to avoid cross-modal prompt isolation.
2. This work designed a dual-objective loss for modality interactions.
3. The experiment results and ablation studies are extensive.

**Weaknesses:**

1. Although being mentioned a lot, there is no justification for the modality bias. It can be good to add some sentences to explain how and why the modality bias will affect model performance.
2. The model performance is only evaluated on two datasets.

**Questions:**

1. Are the Cross-Modal Prompts Query and Cross-Model Prompts Recovery plug-and-play modules? If yes, can they be applied to other models?
2. How were the results generated in Table 12 and 13? Are they generated by replacing one of the query or recovery module with other strategies?

---

### Note · Authors · 2025-11-19

I have read and agree with the venue's withdrawal policy on behalf of myself and my co-authors.